# Tridimensional visualization reveals direct communication between the embryo and glands critical for implantation

Jia Yuan[1], Wenbo Deng[1], Jeeyeon Cha[2], Xiaofei Sun[1], Jean-Paul Borg [3] & Sudhansu.K. Dey [1]

Embryo implantation is central to pregnancy success. Our previous understanding is limited by studying this phenomenon primarily in two dimensions. Here we employ 3D visualization, revealing that epithelial evaginations that form implantation chambers (crypts) consistently arise with preexisting glands, suggesting direct access of glands to embryos within the chamber. While the lobular domains of the glands become more developed, the ductal regions continue to elongate and progressively stretch following implantation. Using diapausing mice and mice with deletion of the planar cell polarity gene *Vangl2* in uterine epithelial cells, we show that dynamic changes in gland topography depend on implantation-competent blastocysts and planar cell polarity. By transferring blastocyst-size beads preloaded with HB-EGF in pseudopregnant mice, we found that HB-EGF is a trigger for the communication between embryos and glands. Glands directly connecting the crypt encasing the embryo during implantation are therefore fundamental to pregnancy success.

---

[1] Division of Reproductive Sciences, Cincinnati Children's Hospital Medical Center, Cincinnati, OH 45229, USA. [2] Department of Medicine, Vanderbilt University Medical Center, Nashville, TN 37232, USA. [3] Centre de Recherche en Cancérologie de Marseille, Aix-Marseille University UM105, Inst Paoli-Calmettes, UMR7258 CNRS, U1068 INSERM, Cell Polarity, Cell signalling and Cancer - Equipe labellisée Ligue Contre le Cancer, Marseille, France. Jia Yuan and Wenbo Deng contributed equally to this work. Correspondence and requests for materials should be addressed to S.K.D. (email: sk.dey@cchmc.org)

Reciprocal interactions between an implantation-competent blastocyst and the receptive uterus are crucial for implantation[1,2]. Unlike many organs, the adult uterus is plastic and undergoes striking morphological, cellular, and molecular changes during pregnancy. These changes engage an interplay of ovarian hormones, transcription factors, growth factors, morphogens, cytokines, and signaling molecules[1]. Dysregulation of any of these pathways results in implantation failure, or defective implantation, which disseminates adverse ripple effects through the remainder of gestation, compromising pregnancy outcomes[1,3–5]. Blood vessels enter the uterus from the mesometrium, situating the uterus along the mesometrial–antimesometrial (M–AM) axis. Implantation occurs within a specialized crypt (implantation chamber) formed by luminal epithelial (LE) evaginations toward the AM pole[1,6].

Blastocyst apposition and attachment within a crypt occurs in the evening of day 4 of pregnancy (day 1 = vaginal plug) in mice. This event is coincident with increased endometrial vascular permeability at the site of the blastocyst[7]. On day 5, stromal cells surrounding the implantation chamber undergo proliferation and differentiation into decidual cells (decidualization). Decidualization supports embryonic growth and placentation to establish pregnancy. Using conditional uterine deletion of *Vangl2* mice by *Pgr-Cre* driver, we have previously shown that planar cell polarity (PCP) signaling is critical for crypt formation[8]. *Pgr-Cre* driver deletes gene expression in all major progesterone receptor (PGR) expressing uterine cell types (myometrial, stromal, and epithelia), and *Pgr-Cre* is active during uterine development early in postnatal life[9]. Therefore, we could not ascertain whether deletion of *Vangl2* in all uterine compartments was responsible for the observed phenotypes or if they were the result of the subtle defects arising during postnatal uterine development, since gland development begins around day 7 of postnatal period[10,11].

The patterning of LE evaginations during crypt formation is similar to directed morphogenetic movements resulting from PCP signaling, which confer spatial identity, especially during organogenesis[12,13]. Aberrant PCP signaling cause developmental anomalies, including defects in neural tube closure and left–right asymmetry[14–16]. Previous studies using two-dimensional visualization explored the mechanism by which epithelial evaginations form crypts at the AM domain aligned with the glands; however, the display of the elegant topography of crypts and glands during implantation remained unknown until our present work.

Vangl2 (Van Gogh-Like Protein 2) is a core component of the PCP signaling. We found in this study that uterine epithelial-specific *Vangl2* deletion in adult mice by *Ltf-Cre* driver profoundly affects female fertility, despite normal uterine receptivity and initial attachment of the blastocyst within the crypt. However, crypt size and shape were altered. We speculated that 3D visualization of implantation sites would unravel a wealth of previously undiscovered information. Using reporter mice, deep tissue clearing, and antibody staining, tridimensional visualization of the implantation sites shows a spectacular and dynamic display of the implantation process in time. We observed that LE evaginations forming the crypts always emerge with preexisting glands. Gland lobules with long ducts extended beyond the implantation chamber at the AM domain and draining gland secretion directly to the crypt containing the implanting embryo. In contrast, mice with epithelial *Vangl2* deletion show that LE evaginations forming the crypts are shallow with glands that are not extended and well developed. These morphological abnormalities are also reflected in sections of implantation sites, which show oval-shaped smaller crypts, predictive of compromised pregnancy outcomes, as opposed to "spear-shaped" crypts in floxed mice on day 5 of pregnancy. More importantly, a direct communication between the gland and crypt homing the

implanting embryo requires the presence of an implantation-competent blastocysts, but not diapausing blastocysts. We provide evidence that heparin-binding EGF-like growth factor (HB-EGF) is a potential mediator of this gland–embryo communication within the crypt.

## Results

### *Vangl2* deletion in the adult uterine epithelium impedes pregnancy.

Using conditional uterine deletion of *Vangl2* mice by *Pgr*-Cre driver, we previously showed that Vangl2 signaling is critical to pregnancy success[8]. The *Pgr*-Cre driver deletes gene expression in all major uterine cell types with the loss of one allele of PGR[17], and *Pgr*-Cre is active during uterine development early in postnatal life when gland development and maturation occur (Supplementary Figure 1a). Deletion of *Vangl2* in all uterine compartments may have conferred subtle changes in postnatal uterine development, while the loss of one PGR allele may have had an effect on the endocrine landscape to influence the observed phenotype. Therefore, Vangl2's cell type-specific role cannot be delineated using this model.

In contrast, lactoferrin (*Ltf*)-Cre driver is exclusively expressed in the adult estrogen-responsive uterine epithelium[18] and efficiently deletes epithelial *Vangl2* in the adult uterus (Supplementary Figure 1b–f), drastically impeding fertility (Fig. 1a, b). Only 30% of plug-positive deleted mice had live pups as opposed to 81% of floxed (*Vangl2*f/f) females (Fig. 1a), and the litter size of *Vangl2*f/f*Ltf*Cre/+ mice is smaller than *Vangl2*f/f females. Interestingly, uterine receptivity as determined by *Msx1, Lif,* and *Ihh* expression is comparable between *Vangl2*f/f and *Vangl2*f/f*Ltf*Cre/+ uteri on day 4 morning (Fig. 1c). The expression pattern of FOXA2, a transcription factor and critical determinant for gland formation and function, remains unaltered in *Vangl2*f/f and *Vangl2*f/f*Ltf*Cre/+ uteri on day 4 (Fig. 1d). Expression of receptors for progesterone and estrogen (PGR and ESR1, respectively) and primary mediators of hormone action in the uterus are also unaltered in floxed and deleted uteri on day 4 morning (Fig. 1e, f). Histological and biochemical assessment of implantation sites show significant abnormalities: crypt size, shape, and its molecular signatures are remarkably altered in *Vangl2*f/f*Ltf*Cre/+ uteri. The initial uterine blue reaction (vascular permeability) demarcating the implantation site is less intense in the deleted mice than that in the floxed mice on day 5 of pregnancy (Fig. 2a). Crypts are normally "spear-shaped" with a tapered extension of the epithelium, but this signature is lost in deleted mice; instead, deleted mice show oval-shaped (day 5) crypts that are much smaller than those of floxed mice (Fig. 2b). Expression of *Ptgs2*, a molecular marker for the attachment reaction, appears in the crypt epithelium and subepithelial stroma surrounding the blastocyst on day 5. Its expression in *Vangl2*f/f*Ltf*Cre/+ uteri was comparable to that in floxed uteri, albeit with a smaller crypt size in the former (Fig. 2c). *Bmp2* is expressed in subepithelial stromal cells on day 5 around the blastocyst and further expands with the progression of decidualization. We found that *Bmp2* expression is lower in *Vangl2*f/f*Ltf*Cre/+ stroma, suggesting that a cross-talk between the epithelial PCP signaling and stromal Bmp2 is impaired (Fig. 2c). Taken together, these results provide evidence that morphogenetic and cellular features of crypts are significantly altered in the absence of epithelial Vangl2 and that crypt shape could be a predictor of implantation success and pregnancy well being.

### Luminal epithelium shows altered cell polarity in mutant mice.

Ex vivo visualization of day 4 isolated intact luminal epithelia immunostained with antibody to Scribble, another member of the PCP family and phalloidin (F-Actin) shows altered cell polarity

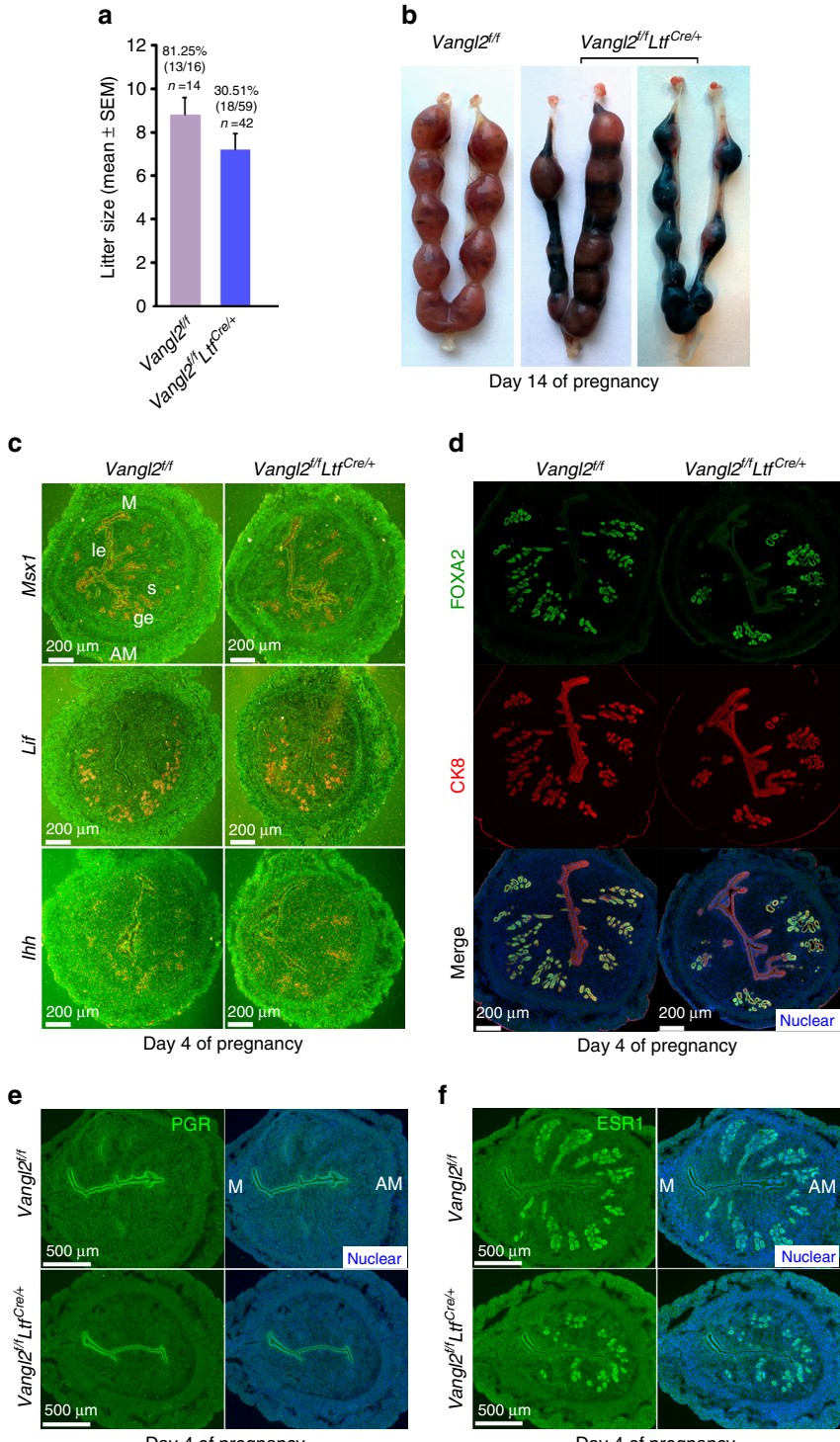

**Fig. 1** *Vangl2* deletion in the uterine epithelium results in severely compromised pregnancy outcome. **a** Percent pregnancy rate and litter size in *Vangl2*f/f and *Vangl2*f/f*Ltf*Cre/+ females. n number of females examined. The number within brackets indicate females with pups over total number of plug-positive females (Mean ± s.e.m. is derived from the indicated number of samples and analyzed by Student's *t*-test). **b** Day 14 pregnant uteri in *Vangl2*f/f and *Vangl2*f/f*Ltf*Cre/+ females. **c** In situ hybridization of *Msx1, Lif, and Ihh* in day 4 pregnant uteri of *Vangl2*f/f and *Vangl2*f/f*Ltf*Cre/+ mice. ge gland epithelium, le luminal epithelium, s stroma, M mesometrial pole, AM antimesometrial pole. Scale bar: 200 μm. **d** IF of FOXA2 and CK8 in day 4 uteri of *Vangl2*f/f and *Vangl2*f/f*Ltf*Cre/+ mice. Scale bar: 200 μm. **e, f** IF of PGR and ESR1 in day 4 uteri of *Vangl2*f/f and *Vangl2*f/f*Ltf*Cre/+ mice. Scale bar: 500 μm. All images are representative of three independent experiments

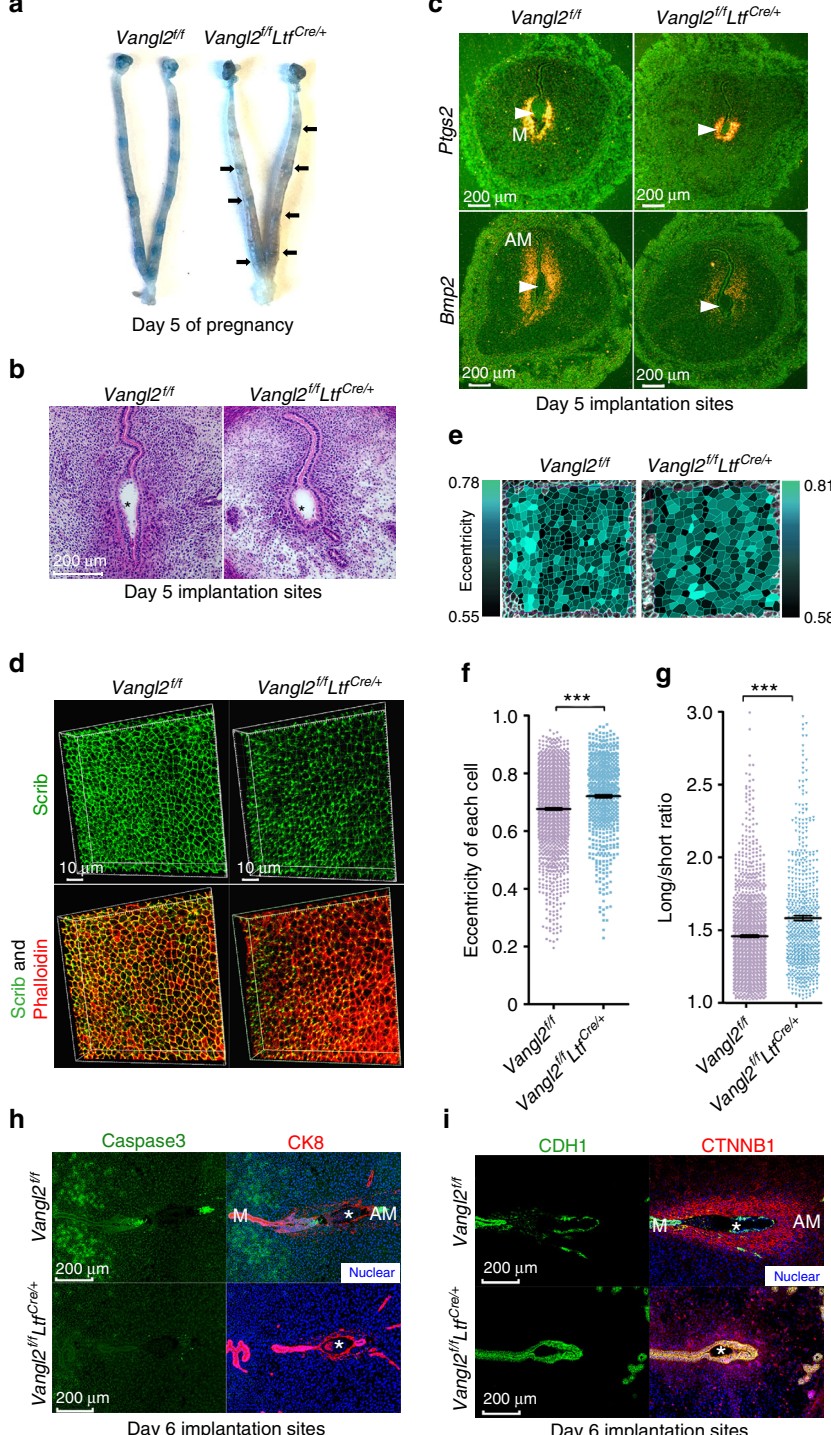

**Fig. 2** *Vangl2* deletion in the uterine epithelium contribute to aberrant implantation and decidualization. **a** Day 5 implantation sites (blue bands) in *Vangl2*<sup>f/f</sup> and *Vangl2*<sup>f/f</sup>*Ltf*<sup>Cre/+</sup> females and arrows indicate weak blue bands. **b** Histology of day 5 implantation sites in *Vangl2*<sup>f/f</sup> and *Vangl2*<sup>f/f</sup>*Ltf*<sup>Cre/+</sup> mice. * embryo. Scale bar: 200 μm. **c** In situ hybridization of *Ptgs2* and *Bmp2* in *Vangl2*<sup>f/f</sup> and *Vangl2*<sup>f/f</sup>*Ltf*<sup>Cre/+</sup> implantation sites on day 5. Scale bar: 200 μm. Arrowheads indicate the location of embryos. **d** 3D reconstruction of Scrib and Phalloidin localization at the apical surface of the epithelium by IF in *Vangl2*<sup>f/f</sup> and *Vangl2*<sup>f/f</sup>*Ltf*<sup>Cre/+</sup> uterine luminal epithelium (LE) on day 4. Scale bar: 10 μm. **e** Cell shape analysis of uterine epithelial cells using SEGGA software. **f, g** Quantification of cell eccentricity and the long-to-short axis ratio in *Vangl2*<sup>f/f</sup> and *Vangl2*<sup>f/f</sup>*Ltf*<sup>Cre</sup> epithelial cells (Mean ± s.e.m. is derived from the indicated number of samples and analyzed by Student's *t*-test, ***$p < 0.001$). **h** IF of Caspase3 and CK8 in day 6 implantation sites of *Vangl2*<sup>f/f</sup> and *Vangl2*<sup>f/f</sup>*Ltf*<sup>Cre/+</sup> mice. Scale bar: 200 μm. **i** IF of CDH1 and CTNNB1 in day 6 implantation sites of *Vangl2*<sup>f/f</sup> and *Vangl2*<sup>f/f</sup>*Ltf*<sup>Cre/+</sup> mice. Scale bar: 200 μm. All images are representative of three independent experiments

(Fig. 2d). To further characterize cell polarity changes in ex vivo intact luminal epithelia of *Vangl2*^f/f and *Vangl2*^f/f*Ltf*^cre/+ mice, a high-throughput quantification of cell characteristics at single-cell resolution was achieved by employing SEGGA image analysis software[19]. We found that LE cells from *Vangl2*^f/f are more isotropic than *Vangl2*-deleted cells based on the eccentricity value of each cell (Fig. 2e, f). The long-to-short axis of each cell was also altered in *Vangl2* deficient cells (Fig. 2e, g). These results show that *Vangl2* deficiency exclusively in LE cells perturb cell shape and organization, which may contribute to crypt shape and size.

**Blastocyst attachment is defective in mutant mice**. Following the attachment reaction, the crypt epithelium in close contact with the blastocyst disappears through entosis[20]. However, the tapered extension of the epithelium at the bottom of the implantation chamber, which is not in direct contact with the blastocyst, shows expression of cleaved Caspase3 signals at midnight of day 5. This expression becomes prominent in almost all epithelial cells in the tapered extension on day 6 morning, resulting in their demise[20]. This epithelial extension perhaps determines the alignment of the crypt axis with the embryonic axis during the early stage of implantation. In contrast, *Vangl2*-deleted crypt epithelium surrounding the blastocyst remains intact and cleaved Caspase3 signals are absent on day 6 (Fig. 2h). These changes are also reflected in the expression patterns of E-Cad (CDH1) and β-Catenin (CTNNB1) co-localization in the crypt epithelium of floxed and deleted mice (Fig. 2i). Subsequently, we found that disturbances at the initial stages of implantation are propagated through later stages of pregnancy, with resorption of implantation sites and poor pregnancy outcomes (Fig. 1b). Although these results provide important information about LE cell features, a deeper and meaningful understanding regarding the interplay between glands and crypts in the context of LE topography requires 3D visualization of intact implantation sites as shown later.

**3D imaging reveals unique embryo–gland communication**. While the two-dimensional view of implantation sites together with genetic, molecular, and physiological approaches have provided important information, the full spectrum of our knowledge on this process remains unfulfilled. Therefore, we sought to explore the complexities of the implantation process through a 3D view in time. Tridimensional visualization of implantation sites unraveled unique aspects of crypt–gland topography and interactions. Although glandular secretions like LIF are essential for embryo implantation in mice, the physical and/or physiological relationship of a crypt with glands is not known. We used tissue clearing with antibody staining and Rosa-tdTomato reporter mice expressed by *Ltf*-Cre driver to create 3D images of implantation sites and glands under multiphoton and light sheet microscopes at different days of pregnancy. We used FOXA2, cytokeratin 8 (CK8), and CDH1 immunostaining to visualize the glands and epithelium. On day 4.5, we observed initiation of epithelial evaginations to form a crypt with glands therein in *Vangl2*^f/f mice. In contrast, these features of crypt formation and glandular display in *Vangl2*^f/f*Ltf*^Cre/+ mice were aberrant and ominous to pregnancy success. Although the glands are already extending and growing in floxed mice, this feature was inferior in *Vangl2*^f/f*Ltf*^Cre/+ uteri (Supplementary Figure 2). When the images of day 5 implantation sites of *Rosa26*^tdTomato*Ltf*^Cre/+ reporter or *Vangl2*^f/f mice immunostained with CDH1 were compared with those from *Vangl2*^f/f*Ltf*^Cre/+ mice, we observed a spectacular crypt–gland landscape. Crypts containing blastocysts are more or less evenly spaced in WT females (Fig. 3a). Interestingly, the gland density in-between the

implantation sites (inter-implantation sites) are much higher due to developing deciduae, which compress the glands at the inter-implantation region (Fig. 3a).

Although gland secretions are essential for implantation, the exact relationship of glands with the embryo within the crypt is not known. Here, we discovered those crypts that contain blastocysts have direct communication with existing glands in WT mice (Fig. 3b and Supplementary Figure 3). More interestingly, multi-lobular glands with long ducts extend past the crypts toward the AM side; glands at the inter-implantation sites are less extended and developed. This pattern is clearly evident in reconstructed segmented and 3D rendered images (Fig. 3b and Supplementary Movie 1). In contrast, *Vangl2*^f/f*Ltf*^Cre/+ females shows irregular epithelial projections and crowded embryos within ill-defined or small crypts with aberrant glandular display (Fig. 3c, d). The gland structures and shapes are withered with poor display of lobules (Fig. 3c, d and Supplementary Movie 2). These results provide evidence that epithelial *Vangl2* signaling is critical for close interactions between glands and crypts encasing the embryo for implantation. Collectively, tridimensional visualization of implantation sites helped to discover unique aspects of embryo–gland topography and communication previously not known.

To further characterize the glands, FOXA2, a gland specific marker, was used for whole-mount staining of day 5 implantation sites in conjunction with CK8, which is expressed in both the luminal and glandular epithelium. The results show that both gland ducts and lobules are FOXA2 positive (Fig. 4a–f). The results clearly show a direct connection of extended glands with the crypt. These results also provide evidence that gland secretions are directly delivered to the crypt homing an embryo, establishing a direct communication between the two entities.

Once the blastocyst attaches within a crypt, the underlying stromal cells undergo proliferation and decidualization. Decidualization is initiated on day 5 with maximal decidualization occurring on day 8. Tridimensional visualization of *Vangl2*^f/f implantation sites on day 6 revealed that emergent ducts connecting the gland lobules from the implantation chambers are stretched further. Most glands are pushed to the periphery, and a few ducts appear severed from the highly lobular glands, perhaps due to pressure from the rapidly growing decidua. Notably, gland density in the inter-implantation sites is higher and ducts are not very-much extended, confirmed by both light sheet and two-photon microscopes (Fig. 5a–f and Supplementary Movie 3). 3D imaging at low magnification of day 6 uterus with implantation sites and inter-implantation sites also support that the glands are compressed in-between implantation sites with numerous less extended glands (Supplementary Figure 4). Interestingly, emergent glands from shallow crypts in *Vangl2*^f/f*Ltf*^Cre/+ have shorter ducts and are shaped like "angel-hair pasta" clustered together (Fig. 5g–i and Supplementary Movie 4). Differences in crypt–gland landscape between *Vangl2*^f/f and *Vangl2*^f/f*Ltf*^Cre/+ are clearly illustrated by fragmentation and 3D image rendering.

**Diapausing blastocysts fail to initiate crypt formation**. On days 4 and 5 of pseudopregnancy during which the ovarian hormone milieu is similar to pregnancy, the lumen appears slightly wavy or flat, and the glands are not well-developed in the absence of embryos (Fig. 6a–f), suggesting that crypt development with the glands necessitates the presence of an active embryo. To assess the importance of embryo during this interaction, delayed implantation model was utilized. In mice and rats, delayed implantation can be initiated by removing the ovaries before the preimplantation estrogen secretion on day 4 morning and

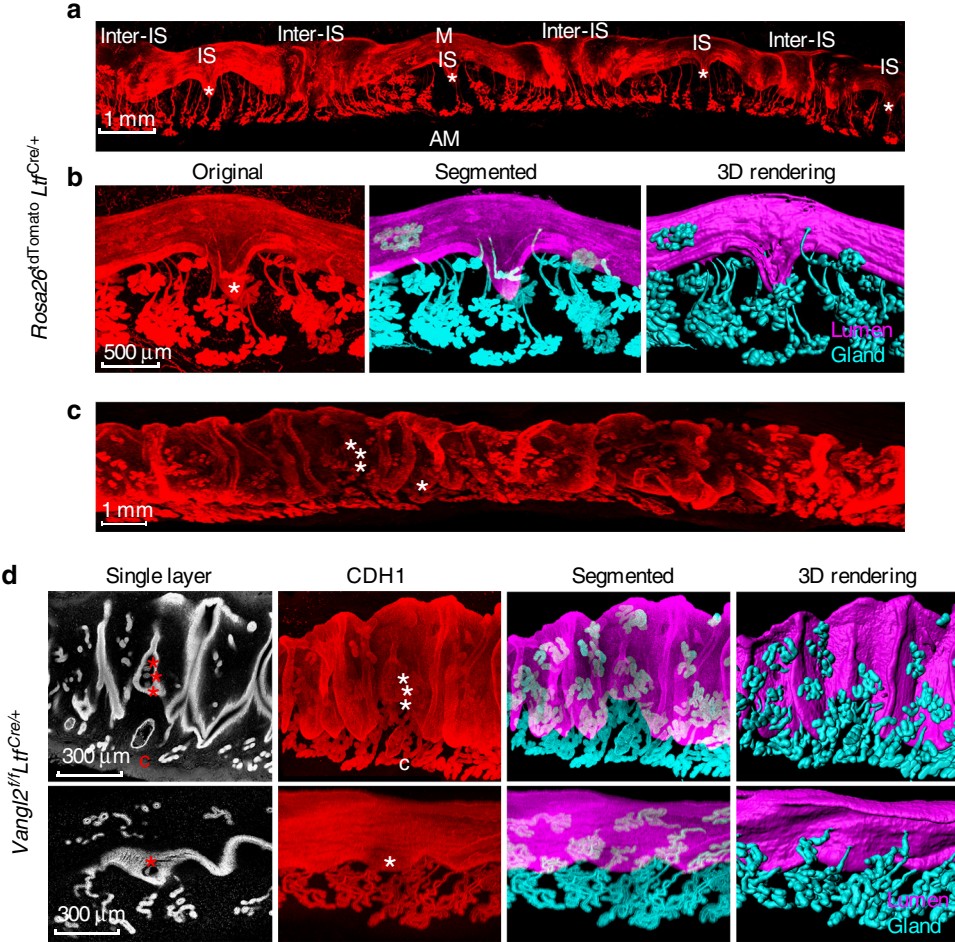

**Fig. 3** 3D images of day 5 implantation sites. **a** Images of one uterine horn in *Rosa26*<sup>tdTomato</sup>*Ltf*<sup>Cre/+</sup> mice on day 5. IS Implantation site, inter-IS inter-implantation site. Images were generated by Nikon A1R Multiphoton Microscope with Plan Apo 4× objective with 50 μm Z-stack. Scale bar: 1 mm. **b** Original images of tdTomato reporter, segmented glands and uterine lumen and 3D rendering of day 5 implantation site in *Rosa26*<sup>tdTomato</sup>*Ltf*<sup>Cre/+</sup> mice. Images were generated by a Nikon A1R Multiphoton Microscope with LWD 16× water objective with 3 μm Z-stack. Scale bar: 500 μm. **c** Image of one uterine horn in *Vangl2*<sup>f/f</sup>*Ltf*<sup>Cre/+</sup> mice on day 5 immunostained with E-cad (CDH1). Images were generated by Nikon A1R Multiphoton Microscope; Plan Apo 4× objective with 50 μm Z-stack. Scale bar: 500 μm. **d** Single layer, whole-mount staining, segmented and 3D rendering of images in day 5 implantation site in two independent *Vangl2*<sup>f/f</sup>*Ltf*<sup>Cre/+</sup> mice immunostained with E-cad (CDH1). Images were generated by Nikon A1R Multiphoton Microscope; LWD 16× water objective with 3 μm Z-stack. * embryos, C cyst. Scale bar: 500 μm. All images are representative of three independent experiments

maintaining them on daily $P_4$ administration. Under this condition, blastocysts undergo dormancy (diapause) in the quiescent uterus, and the process of implantation remains in suspended animation. However, implantation can be initiated in these $P_4$-primed uteri by injecting a small amount of estrogen within 15–20h[21,22]. 3D imaging showed that diapausing blastocysts cannot trigger crypts and glands fail to extend (Fig. 6g–j), as opposed to well-formed crypt with extended ducts and lobules following termination of the delayed implantation with blastocyst activation by estrogen (Fig. 6k–n). Interestingly, receptive uteri in the absence of blastocysts exhibit a similar scenario as delayed uteri with diapausing blastocysts (Supplementary Figure 5a–c); only activated embryos can trigger appropriate communication (Supplementary Figure 5d–f). These results suggest that implantation-competent blastocysts initiate crypt formation and is critical for direct communication with the glands within the crypts in the receptive uterus. We next asked what could be a potential linker of this event.

**HB-EGF is a mediator of embryo–gland communication.** We have previously shown that HB-EGF is the earliest known molecular marker that appears exclusively in the crypt epithelium at the time of implantation, but the expression is absent in the delayed uterus at the site of blastocysts[23]. We also found that implantation-competent (activated) blastocysts express *Hbegf*, and blastocyst-size Affi-gel blue beads carrying HB-EGF can trigger implantation-like responses if transferred into receptive pseudopregnant mouse uteri on day 4 morning. Furthermore, HB-EGF can induce its own gene *Hbegf* in the uterus[24,25]. These results suggest that HB-EGF originating in the active blastocyst triggers its own gene in the crypt epithelium and elicits a direct communication between the blastocysts and glands within the developing crypts similar to that seen in normal implantation. Our 3D visualization shows that HB-EGF-carrying beads when transferred into day 4 pseudopregnant uteri exhibit similar gland–crypt responses on day 5 (Fig. 7a–d). Control beads carrying BSA failed to show such effect (Fig. 7e–h). This finding is consistent with the expression of HB-EGF, exclusively, in the crypt epithelium surrounding the blastocyst (Fig. 7i). These results provide evidence that HB-EGF is at least one of the potential mediators for the critical interaction between the glands and crypts homing the embryos.

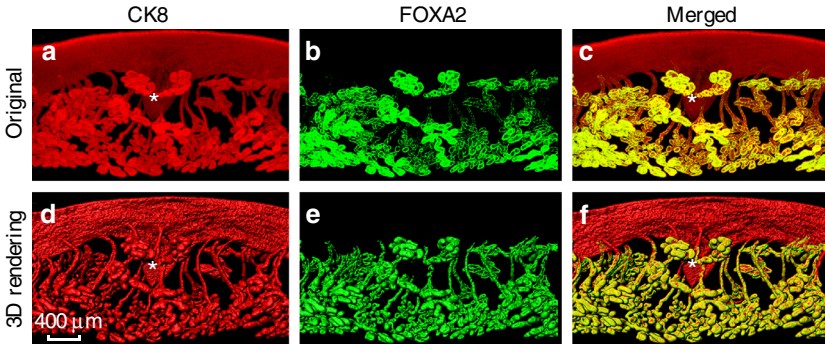

**Fig. 4** 3D visualization of day 5 implantation sites in *Vangl2*[f/f] mice stained with CK8 amd FOXA2. **a–f** Original double immunostaining with CK8 and FOXA2 and 3D rendering pictures. CK8 markes both luminal epithelial and glandular epithelial cells, while FOXA2 only marks glands. Representative images of three independent experiments, images were generated by Nikon A1R Multiphoton Microscope; LWD 16× water objective with 3 μm Z-stack. * location of embryos. Scale bar: 400 μm

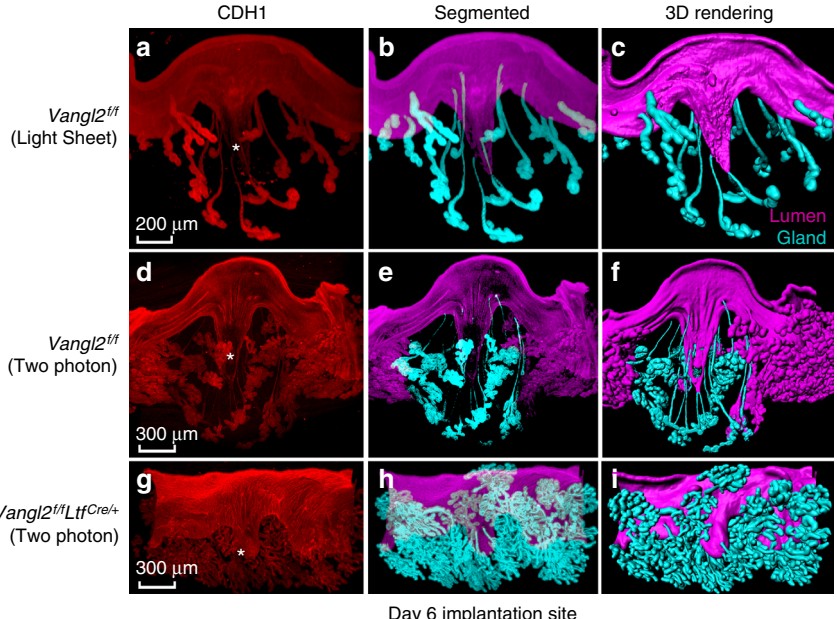

**Fig. 5** 3D visualization of day 6 implantation sites in *Vangl2*[f/f] and *Vangl2*[f/f]*Ltf*[Cre/+] mice. **a–c** 3D images of CDH1, segmented and 3D rendered images of *Vangl2*[f/f] day 6 implantation sites generated by a light sheet microscopy. Scale bar: 200 μm. **d–i** 3D immunostaining with CDH1, segmented and 3D rendered images, respectively, in *Vangl2*[f/f] and *Vangl2*[f/f]*Ltf*[Cre/+] mice generated by two-photon microscopy. Images are generated by Nikon A1R Multiphoton Microscope with a LWD 16× water objective with 3 μm Z-stack. Note: aberrant crypt size and evagination in *Vangl2*[f/f]*Ltf*[Cre/+] mice with derailed extension and display of glands. * location of embryos. Scale bar: 300 μm. All images are representative of three independent experiments

**Poor decidual and embryo growth in *Vangl2*[f/f]*Ltf*[iCre/+] mice.** Tridimensional imaging after CDH1 localization revealed several unique features in day 8 WT mice. The main lumen at the mesometrial side becomes occluded and then splits into a squamous layer and a cuboidal layer of epithelia to form a secondary lumen[26]. At this time, many ducts connecting the gland lobules at the implantation chamber are further stretched toward the AM pole and become disconnected, perhaps due to shear stress secondary to expanding decidua and implantation chamber; most ductless glands remain situated at the periphery of the AM decidua (undifferentiated stroma just above the myometrium). The glands at the inter-implantation sites remain clustered together due to pressure from the expanding implantation chambers (Fig. 8a–c and Supplementary Movie 5). This scenario is greatly compromised in *Vangl2*[f/f]*Ltf*[Cre/+] mice on day 8. Many ducts from the implantation chambers are still attached to the gland lobules due to reduced growth of the decidua with impaired embryonic growth. The implantation site is also more roundish compared with oval implantation site in *Vangl2*[f/f] mice (Fig. 8d–f and Supplementary Movie 6).

To better understand the functionality of glands in the implantation sites and inter-implantation sites on day 8, we used longitudinal sections of two implantation sites joint together and found that glands at the implantation sites show little or no expression of FOXA2 and ESR1, and absence of *Lif* (Supplementary Figure 6a). In contrast, the glands at the inter-implantation site were positive for FOXA2 and ESR1 expression. Although it is known that *Lif* expression in the gland is transient during uterine receptivity primarily on day 4, expressions of FOXA2 and ESR1 at lower (Supplementary Figure 6b) and higher magnifications (Fig. 8g) are more persistent with the exception of ductless glands in implantation sites on day 8. To explore the relationship and function of FOXA2 and ESR1 in uterine glands, we performed co-immunoprecipitation (IP) experiments, since there is evidence that FOXA2 interacts with ESR1 in the liver[27]. Protein–protein interaction between ESR1 and FOXA2 was confirmed by IP

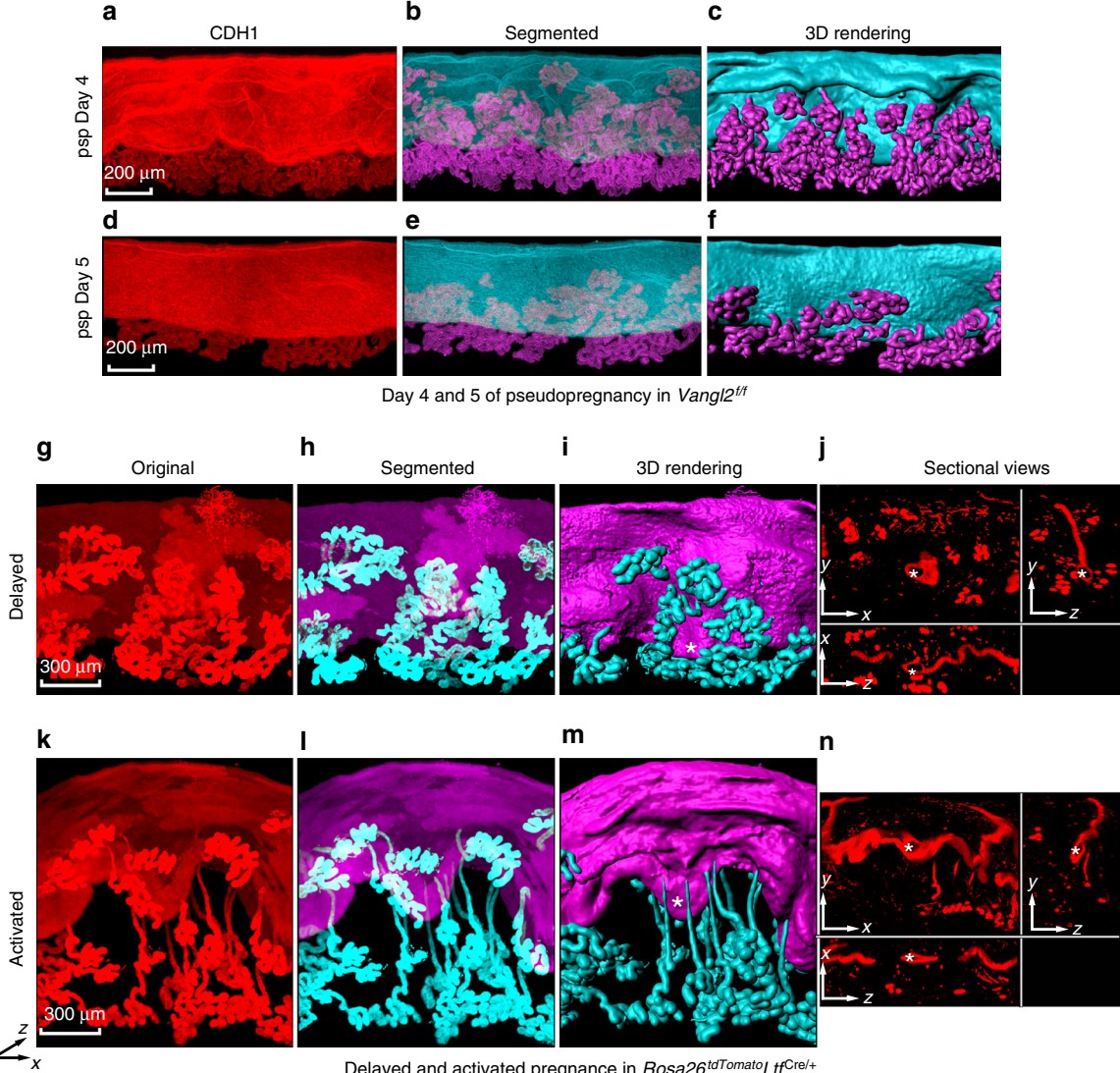

**Fig. 6** Implantation-competent blastocysts are required for crypt formation. **a–f** IF of E-Cad (CDH1), segmented and 3D rendering, respectively, in pseudopregnant days 4 and 5 uteri. psp pseudopregnancy. Scale bar: 200 μm. **g–n**, Original images of tdTomato reporter, segmented glands, and uterine lumen, 3D rendering and sectional view of implantation sites at 24 h after terminating delayed implantation by an injection of estrogen in $P_4$-primed $Rosa26^{tdTomato}Ltf^{Cre/+}$ mice. Asterisks indicate locations of embryos in delayed and activated uterus. Scale bar: 300 μm. Sectional views (**j**, **n**) are represented images (**g**, **k**). Images are generated by Nikon A1R Multiphoton Microscope with a LWD 16× water objective with 3 μm Z-stack. All images are representative of three independent experiments

experiments (Supplementary Figure 6c). These observations suggest that the glands at the implantation site become nonfunctional if their communication from the implantation chamber is cut off. Glands that are present in the inter-implantation sites may carry out necessary functions for pregnancy progression. It has been proposed that glands are necessary for maintaining pregnancy past implantation in mice[28,29], supporting our observation of persistent presence of functional glands past implantation at least until day 8 at inter-implantation sites. This suggests that direct communication between the blastocyst and glands within the crypt is critical during the early stages of implantation. There is now evidence that glands in the human endometrium start declining after the first trimester of pregnancy[30].

Embryonic growth is halted in $Vangl2^{f/f}Ltf^{Cre/+}$ mice as seen on day 8. From this stage forward, many embryos start showing resorption, and by day 14 they are severely damaged with virtually no placental remains (Fig. 1b). Collectively, this study

highlights the unique and dynamic spatial relationship between crypt formation and glandular display for implantation, decid-ualization, and pregnancy success. Although uterine receptivity is critical for blastocyst attachment and implantation, epithelial-specific Vangl2/PCP signaling in conjunction with an activated blastocyst and glands in the crypt plays a critical yet previously unrecognized role in implantation success.

## Discussion

The invariable LE evagination to form a crypt with the existing glands, setting up a direct communication between the blastocyst and glands within the crypt, is a previously unknown event in implantation biology. The finding that this feature becomes aberrant in epithelial-specific Vangl2-deleted uteri despite the attainment of receptivity is a testament to epithelial Vangl2/PCP's role in the organization of glands with crypts. Two-dimensional imaging of uterine sections probed with molecular markers and

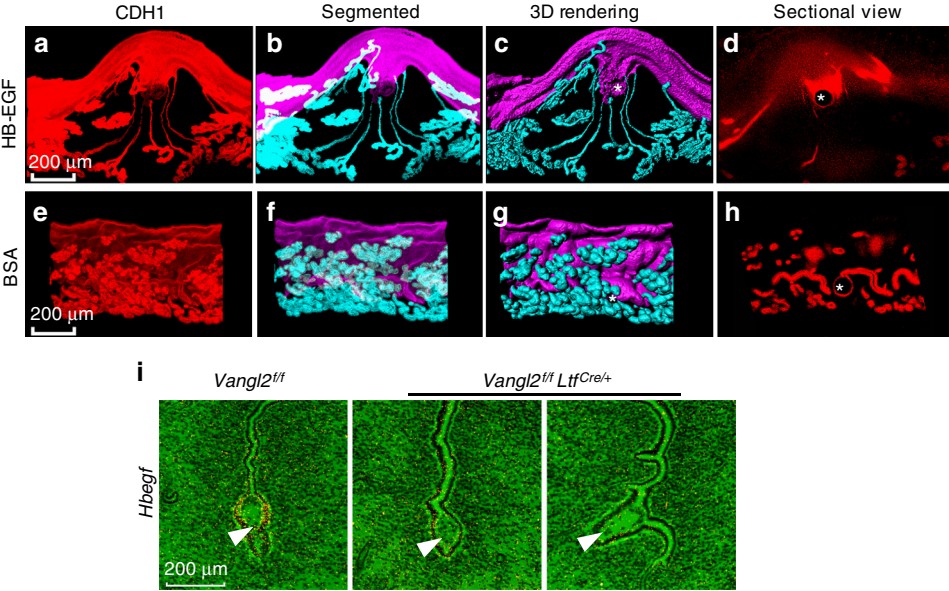

**Fig. 7** 3D analysis of day 6 crypt induced by transfer of Affi-gel blue beads presoaked in HB-EGF or BSA. **a–d** IF of E-Cad (CDH1), segmented glands, and uterine lumen, 3D rendering and sectional view of day 6 crypts at the site of beads carrying HB-EGF transferred into uteri of *Vangl2*^f/f mice. **e–h** Original image, segmented glands, and uterine lumen, 3D rendering and sectional view of day 6 at the site of beads carrying BSA uteri of *Vangl2*^f/f mice. Asterisks indicate the location of beads. Scale bar: 200 μm. Images are generated by Nikon A1R Multiphoton Microscope with a LWD 16× water objective with 3 μm Z-stack. **i** In situ hybridization of *Hbegf* in *Vangl2*^f/f and *Vangl2*^f/f*Ltf*^Cre/+ implantation sites on day 5. Arrowhead indicates embryo. Note: absence of *Hbegf* expression in the crypt epithelium. Scale bar: 200 μm. All images are representative of three independent experiments

histological experiments provide evidence for the essential role of crypt shape, size, and structure in implantation and pregnancy success, but the complex cellular configuration of the implantation process in time is far from our present comprehension. In this context, tridimensional visualization of intact implantation sites has provided a wealth of new information and depicted a unique landscape. The coexistence of the embryo and glands in the crypt (implantation chamber) and their growth and migration toward the AM domain is an unprecedented finding that will open the door for many questions. Aberration of these events, captured in a 3D context in *Vangl2*-deleted epithelia, provides compounding evidence for the role of epithelial Vangl2/PCP signaling in implantation. However, Vangl2/PCP signaling primarily becomes active in the presence of an implanting embryo in collaboration with HB-EGF in the uterus very-early in pregnancy.

The essential role of glands in implantation has long been recognized[31–33]. Although it is thought that glandular secretions are delivered to the lumen and/or stroma surrounding the glands, our finding that the glands secrete directly into the implantation chamber (crypt) containing the embryo has not been previously described. This information unveils a new avenue of investigation begging the question of optimal number of glands needed to be linked to a crypt for implantation and pregnancy success. It could be argued that the gland density in a given area of the LE determines the site of crypt formation. It seems that fewer, highly lobular glands are associated with crypt formation, which is not seen in *Vangl2*-deleted pregnant mice with inferior crypt structure and underdeveloped glands. It is conceivable that a cross-talk between the embryo and glands drives this phenotype. The delayed implanting mice primed with $P_4$ fail to show this embryo–gland interaction, suggesting that the quiescent uteri and diapausing blastocysts cannot evoke this response. In contrast, initiation of blastocyst activation by terminating delayed implantation with an injection of estrogen in $P_4$-primed delayed mice triggers blastocyst-gland cross-talk within 24 h. The absence of similar gland–crypt topography in day 4 and day 5 of pseudopregnant mice that do not have embryos with comparable

ovarian hormone levels to those in pregnant mice on these days further reinforces our hypothesis that changing ovarian steroid hormone levels do not appear to influence dynamic crypt–gland topography and interaction seen in pregnancy in time; rather these events depend on implanting embryos. How this cross-talk between glands and crypts homing embryos is executed is a critical question. One potential mediator is HB-EGF. We have previously shown that this growth factor can function in a paracrine and/or juxtacrine manner and is expressed in both the activated blastocyst and crypt epithelium during implantation; however, *Hbegf* is not expressed in the delayed uterus or diapausing blastocysts[23–25,34]. HB-EGF can induce its own gene *Hbegf*, and blastocyst-size beads presoaked in HB-EGF can elicit implantation-like responses in the receptive uterus similar to those elicited by the living blastocysts during normal implantation[24]. These results suggest that HB-EGF is a potential player that establishes a hierarchy of events between the embryo and crypt epithelium, which first originates from an implantation-competent blastocyst. This auto-induction loop of HB-EGF thus establishes a molecular bridge between these two different entities for successful implantation. These features of crypt–gland topography suggest that Vangl2/PCP active during the process of implantation is critical for pregnancy success, but this process requires collaboration with an implantation-competent (activated) blastocyst and HB-EGF as partners.

It is generally believed that the glands are destroyed during pregnancy[31]. Surprisingly, our findings show the glands' presence continues at least through day 8 of pregnancy. In histological analysis of cross-sections of day 8 implantation sites in 2D imaging, the presence of glands may often not be evident, since they were pushed toward the AM pole close to the myometrium due to decidual growth and expansion of the implantation chamber. Tridimensional imaging shows that long ducts connecting the highly lobular glands are stretched starting from day 5. These long and slender ducts begin to show signs of breaching from day 6, with duct-severed glands clearly evident on day 8, which raises interesting questions about the fate of these ductless glands. The

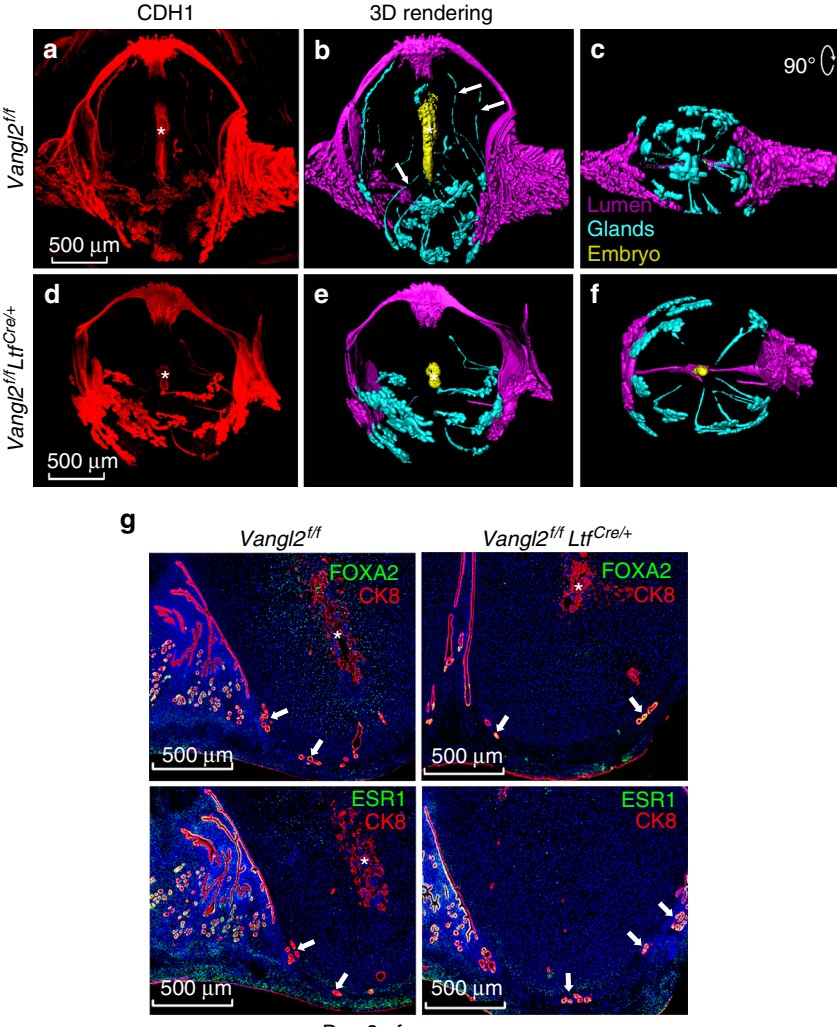

**Fig. 8** 3D analysis of day 8 implantation sites in *Vangl2*^f/f and *Vangl2*^f/f*Ltf*^Cre/+ mice. **a, b, d**, and **e** Images of CDH1 immunostained and 3D rendered images of day 8 implantation sites in *Vangl2*^f/f and *Vangl2*^f/f*Ltf*^Cre/+ mice, respectively, generated by two-photon microscopy. Arrows show the sheared ducts from the gland lobules. **c, f** Images correspond to a clockwise rotation of 90° from the bottom view of *Vangl2*^f/f and *Vangl2*^f/f*Ltf*^Cre/+ implantation site. Images were generated by a Nikon A1R Multiphoton Microscope with Plan Apo λ 10× objective with 12 µm Z-stack. **g** IF of FOXA2 with CK8 and IF of ESR1 with CK8 on day 8 implantation sites in V*angl2*^f/f and *Vangl2*^f/f*Ltf*^Cre/+ mice. Arrows indicate glands at the implantation site. * location of embryos. Scale bar: 500 µm. All images are representative of three independent experiments

absence or very-weak expressions of FOXA2 and ESR1 with no *Lif* expression in these glands suggest their gradual demise or nonfunctionality. In contrast, most glands at the inter-implantation site were FOXA2 and ESR1 positive, but *Lif* negative. These observations suggest that the glands at the implantation site become dysfunctional if their communication from the implantation chamber is cut off. On the other hand, the glands that are present in inter-implantation sites may carry out necessary functions for pregnancy progression. Indeed, a recent study suggests that factors other than LIF from these glands may help pregnancy to progress by nourishing the placenta[28]. There is now evidence that glands in the human endometrium start declining after the first trimester[30].

Additionally, we observed the formation of a bilaminar secondary lumen forming from the occluded lumen at the mesometrial domain near the inter-implantation site that encases decidua as seen on day 8 implantation sites is an important finding. The occlusion and reformation of the uterine lumen during pregnancy was reported more than three decades ago in rats[26]. The restructuring and migration of the bilaminar edges of

the epithelia toward the mesometrial decidua are connected with bilaminar epithelia at the lateral sites of the inter-implantation regions on both sides of the implantation chamber. The epithelial sheet immediately underneath the decidua progressively degrades along with the sequential degeneration and regeneration of the epithelial sheet at the mesometrial site, ultimately encircling the embryo and extra-embryonic membranes within the lumen to prepare for parturition[26]. Observing similar events in mice through 3D visualization provides support and expansion of this concept. These findings argue against the regeneration of the LE from stromal stem cells in pregnancy[35], since the lumen is formed from the existing LE sheets. The molecular players in restructuring and migration of the bilaminar epithelium are yet to be determined through careful long-term studies.

Another important question is the importance of the "spear-shaped" crypt seen in normal day 5 implantation sites as opposed to "oval-shaped" crypt in deleted mice. We argue that the tapered tip in the normal crypt serves as a guide to align the embryonic axis with the implantation chamber axis even at this early stage. Caspase 3 activation at the tip on day 6 when the implantation

process is well underway suggests its demise; its presence is no longer required, because the embryonic axis is already aligned with the implantation chamber axis. This is a critical juncture in pregnancy establishment. In *Vangl2*-deleted mice, the defective crypt shape and size perpetuates adverse effects throughout the course of pregnancy, establishing epithelial PCP's critical role in early pregnancy. Changes in cell shape, rotation of angle, and its eccentricity seen in ex vivo epithelial explants of *Vangl2*-deleted uteri further support this conclusion.

Our findings show that gland ducts connecting the embryo within the crypt are extended with their developing lobules. A recent study found that glands are oriented toward the implanting embryo[36]. We believe that this appearance is due to passive stretching of the glands from the crypt by developing decidua. In conclusion, our tridimensional visualization shows that crypt formation from the LE consistently occurs with preexisting glands and that dynamic changes in time are coordinated. It is conjectured that epithelial Vangl2/PCP signaling becomes active in the adult uterus in collaboration with the implantation-competent blastocyst and HB-EGF and sets the stage for pregnancy success very-early during crypt formation. While there has been a great deal of attention on endometrial receptivity for human implantation, the present findings will provoke interest on the role of glands and their interactions with the growing embryo throughout this process. It would be interesting to see if implantation of the embryo in close proximity with glands is common to other species including humans. Indeed, a report shows a conceptus from the earliest human specimen (~43 days of menstrual age) available implanted in the superficial layer of the secretory endometrium overlying well-developed endometrial glands[30]. Further studies utilizing 3D visualization of implantation chambers in humans, subhuman primates, and large animals may elucidate the conserved nature of gland–embryo interaction across species.

## Methods

**Mice.** *Vangl2*$^{f/f}$*Ltf*$^{Cre/+}$ mice were generated by mating *Vangl2* floxed (*Vangl2*$^{f/f}$, C57BL/6,129 mixed background) females[15] with *Ltf*$^{Cre/+}$ males (C57BL/6,129 and albino B6 mixed background)[18]. *Rosa26*$^{tdTomato}$*Ltf*$^{Cre/+}$ mice were generated by mating *Ltf*$^{cre/cre}$ males with *Rosa26*$^{tdTomato}$ female mice (C57BL/6 J Jackson lab). *Vangl2*$^{f/f}$*Pgr*$^{Cre/+}$ mice were generated by mating *Vangl2* floxed (*Vangl2*$^{f/f}$, C57BL/6,129 mixed background) female with *Pgr*$^{Cre}$ males[17]. All mice used in this study were housed in a temperature-controlled room under a constant 12 h/12 h light/dark cycle in the Cincinnati Children's Animal Care Facility according to NIH and institutional guidelines for the use and care of laboratory animals. The females and males were randomly bred and mated. All protocols were approved by the Cincinnati Children's Animal Care and Use Committee. Mice were provided with autoclaved Laboratory Rodent Diet 5010 (Purina) and UV light-sterilized reverse osmosis/deionized constant-circulation water ad libitum. At least three mice from each genotype were used for individual experiment.

**Analysis of pregnancy events.** Three adult females from each genotype were randomly chosen and housed with a WT fertile male overnight in separate cages; the morning of finding the presence of a vaginal plug was considered successful mating (day 1 of pregnancy). WT and mutant plug-positive females were then housed separately until processed for experiments. After three consecutive nights of co-habitation, the male and females were separated and rested for several days. Plug-positive females were selected for pregnancy experiments. Litter size, pregnancy rate, and outcomes were monitored in timed pregnancy. Blue reaction was performed by injecting intravenously a blue dye solution (Chicago Blue dye) 3–4 min before the mice were killed. The distinct blue bands along the uterus indicated the site of implantation[3,5,8]. For confirmation of pregnancy in plug-positive day 4 mice or mice showing no blue bands on day 5, one uterine horn was flushed with saline for the presence of blastocyst. If blastocysts were present, the contralateral horn was used for experiments and mice without any blastocyst were discarded. Pseudopregnancy was induced by mating females with vasectomized males. To induce delayed implantation, pregnant mice on day 4 morning were ovariectomized before the preimplantation estrogen secretion and injected with progesterone (P$_4$, 2 mg/mouse) from days 5 to 7 and were killed on day 8. For activation of blastocyst with termination of delay, estradiol-17β (E2, 25 ng) and P4 (2 mg/mouse) were injected on day 7 of delay. Mice were killed on day 8 (equivalent to day 5 of normal implantation) and subjected to 3D imaging[21,22].

**Transfer of protein-carrying beads.** Briefly, embryo-size Affi-Gel Blue beads (Bio-Rad; 100–200 mesh, no. 153–7302) were washed 6 times with sterile PBS and then incubated with 20 μl rhHB-EGF (100 ng/μl, R&D, Cat. 259-HE) or ultrapure BSA (100 ng/μl, Invitrogen™, Cat. AM2616) in 37 °C for 1 h. Beads were briefly washed 4 times in sterile PBS before transfer into uteri (7 beads/uterine horn of day 4 pseudopregnant CD1 or *Rosa26*$^{tdTomato}$*Ltf*$^{Cre/+}$ mice on day 4 of pseudo-pregnancy produced by mating females with sterile vasectomized males. Mice were killed 48 h later. Implantation-like responses were demarcated by blue dye injection[24].

**Immunofluorescence (IF).** IF for ESR1 (1:300, sc-542; Santa Cruz), CDH1 (1:300, sc-31020, Santa Cruz), Scribble (1:1000, H300, Santa Cruz), CTNNB1 (1:500, sc-1496, Santa Cruz), PGR (1:300, 8757; Cell Signaling technology), rabbit CDH1 (1:300, 3195s, Cell Signaling Technology), Vangl2 (1:10,000, custom made in JP Borg's laboratory, INSERM), ZO-1 (1:300, 33–9100, Invitrogen), phalloidin (1:1000, 43166A, Thermo Scientific), FOXA2 (1:300, WRAB-FOXA2, Seven Hills Bioreagents), CK8 (1:1000, TROMA-1, Hybridoma Bank, Iowa), and Caspase 3 (1:300, 9661s, Cell Signaling technology) was performed using secondary antibodies conjugated with Cy-2, Cy-3, Alexa 488, or Alexa 594 (1:300, Jackson Immuno Research). Nuclear staining was performed using Hoechst 33342 (1:500, H1399, Thermo Scientific). Tissue sections from control and experimental groups were processed onto the same slide.

**Whole-mount immunostaining of luminal epithelium sheet (LE).** IF was performed in separated LE[8]. In brief, day 4 intact luminal epithelia were separated by mild enzymatic treatment and subjected to immunofluorescence with an antibody to Scribble, a member in the PCP family, and phalloidin (F-actin). Imaris software was applied to create 3D reconstructions of images[8].

**Single-cell analysis of epithelial cell divergence.** SEGGA software was employed to analyze cell shape at single-cell resolution[19]. Cell eccentricity and the long-to-short axis ratio of each cell were analyzed in ex vivo epithelial sheets from *Vangl2*$^{f/f}$ and *Vangl2*$^{f/f}$*Ltf*$^{Cre/+}$ mice. Cell eccentricity, a measure of cell elongation, is the square root of 1 minus the squared ratio of the short axis length to the long axis length of an ellipse fit to each cell. An isotropic, circular cell yields a value of zero, while highly anisotropic cells yield values approaching 1.

**RNA isolation and qPCR.** RNA was isolated and reverse transcribed into cDNA and quantitative RT-PCR was performed in ABI Thermal Cyclers Real-Time PCR machine[3]. The primers sequence of *Vangl2* and *Rpl7* are: 5′-CCAAA-CAGCAGCCTTACCAT-3′ and 5′-TCCTTAAGCCCATTGGTCAC-3′ for Vangl2 (product size 196 bp); and 5′-GCAGATGTACCGCACTGAGATTC-3′ and 5′-ACCTTTGGGGCTTACTCCATTGATA-3′ for mouse *rpL7* (product size, 129 bp)[8,37]. *rpL7* served as an internal control and the relative expression was calculated according to manufacturer's protocol.

**Immunoblotting.** Tissues were homogenized in RIPA buffer with protease and phosphatase inhibitors. The protein extracts were run in 10% Bis-acrylamide gel and transferred to PVDF membranes. After blocking in 5% non-fat milk, the membranes were probed with Vangl2 (1:1000) and Actin (1:1000, SC-1615, Santa Cruz) overnight in 4 C. The membranes were washed in TBS-T and incubated with HRP-conjugated secondary antibody and visualized with an ECL kit (RPN2108; GE Healthcare)[8]. The scans of uncropped films of western blotting are shown in Supplementary figure 7 (panels a, b and c).

**In situ hybridization.** The $^{35}$S-labeled probes were used for in situ hybridization[3,6,23]. Control and experimental sections were mounted onto the same slide, prehybridized, and hybridized in formamide buffer for 4 h at 45°C. The slides were then coated with Carestream NTB emulsion (Carestream, Cat no. 889 5666) and autoradiographs were developed and visualized under a Nikon dark-field microscope. Mouse-specific anti-sense cRNA probes for *Msx1*, *Ihh*, *Cox2*, *Lif*, and *Hbegf* were used.

**Co-immunoprecipitation assays.** Co-IP was performed by homogenizing day 4 pregnant uterine tissues in RIPA buffer (10 mM HEPES, 142.5 mM KCl, 5 mM MgCl2, 1 mM EGTA, 0.5% NP-40, pH 7.5) containing protease inhibitor cocktail (Roche, 04693159001). Protein (2 mg) for each Co-IP sample was incubated with 1 μg of specific antibody and 5 μl Protein G Dynabeads (Invitrogen, 10003D) overnight at 4 °C. Collected beads were washed 5 times by RIPA buffer. The samples were immunoblotted by specific antibodies[38].

**Whole-mount immunostaining for 3D imaging.** This method was modified and combined according to published paper[39]. Samples were fixed in Dent's Fixative (Methanol:DMSO (4:1)) overnight in −20 °C and then washed with 100% methanol[36,40]. The samples were bleached with 3% H$_2$O$_2$ in methanol at 4 °C overnight to remove pigmentation. After washing in PBS-T containing 0.2% gelatin (PBST-G) for 6 times with 1 hour each, samples were incubated with indicated

antibodies (E-cad: 1:100, FoxA2: 1:100, CK8: 1:100) at room temperature in a rotor for 7 days. After incubation, the samples were washed by PBST-G six times for 1 hour each and incubated with Alexa-conjugated secondary antibodies in light-proof box for 4 days at room temperature. After six washes in PBST-G at room temperature, the samples were stored in the dark until tissue clearing.

**Tissue clearing**. A modified-3DISCO method was applied for tissue clearing[39]. Briefly, the stained samples were incubated in 50% tetrahydrofuran (THF; Sigma-Aldrich) overnight then dehydrated in 80% and 100% THF for 1 hour each. Delipidation by dichloromethane (DCM; Sigma-Aldrich) for 30 min was followed by overnight clearing step in dibenzyl ether (DBE; Sigma-Aldrich)[41]. For $Rosa26^{tdTomato}Ltf^{cre/+}$ mice, uteri were fixed in paraformaldehyde (PFA), the muscle layer partially removed by a pair of corneal scissors, and then clearing in RIMS (Refractive Index Matching Solution) overnight[42].

**3D imaging and processing**. 3D image acquisitions were performed by using a Nikon multiphoton upright confocal microscope (Nikon A1R) or light sheet (ZEISS Z.1). Samples were laid on slides, covered with DEB or RIMS, and enclosed by cover slips for multiphoton confocal imaging using 4× (Plan Apo, 50 μm Z-stack), 10× (Plan Apo λ, 12 μm Z-stack) air objective for low magnification and 16× (LWD, 3 μm Z-stacks) water objective for high magnification. The samples were immersed in RIMS for light sheet confocal imagings using Lightsheet Z.1 5× detection optics. All files generated by Nikon element and ZEN were imported into Imaris (version 8.3, Bitplane) for visualization and 3D reconstruction. To obtain the 3D structure of the tissue, the surface tool was utilized for 3D rendering visualizations. To isolate a specific region of the tissue, the surface tool was manually used to segment the images and the mask option was selected for subsequent pseudo-coloring. 3D images and movies were generated using the "snapshot" and "animation" tools[39].

**Statistics**. Each experiment was repeated several times depending on the consistency of the results. Statistical analyses were performed using a two-tailed Student's $t$-test. A value of $p < 0.05$ was considered statistically significant.

**Data availability**. The authors declare that all data supporting the findings of this study are available within the article and its Supplementary Information files or from the corresponding authors on reasonable request.

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

## Acknowledgements

We sincerely appreciate Katie Gerhardt's time and effort in editing the manuscript. We are grateful to Alain Chedotal (INSERM, France) for his help in 3D imaging and to Jennifer Zallen (Memorial Sloan Kettering Cancer Center, New York, NY, USA) for her help in the use of the SEGGA software for rapid image analysis of epithelial cells. We also thank Matt Kofron for his excellent support and help for confocal imaging at our institutional Nikon Center of Excellence. *Pgr-Cre* mice were original obtained from Francesco DeMayo and John P Lydon (Baylor medical center). Yingzi Yang (Harvard) originally provided the floxed Vangl2 mouse line. This work was supported in parts by NIH grants (R01HD068524, DA006668, and P01CA77839), and grants from the March of Dimes (to S.K. D.). J.C. was a National Research Service Award fellow (F30AG040858) of the University of Cincinnati Medical Scientist Training Program. J-P.B.'s laboratory is funded by La Ligue Nationale Contre le Cancer SIRIC (INCa-DGOS-Inserm 6038), and Ruban Rose. J-P.B. is a scholar of Institut Universitaire de France.

## Author contributions

J.Y., W.D., J.C., and X.S. performed experiments, J.Y., W.D., and S.K.D. designed experiments. J-P.B. provided valuable reagents and tools. J.Y., W.D., J.C., X.S., J-P.B., and S.K.D. analyzed data. Y.J., W.D., J.C., and S.K.D wrote the manuscript.

## Additional information

**Competing interests:** The authors declare no competing financial interests.

