## [Peer Review File · Nature Communications]

Reviewers' comments:

Reviewer #1 (Remarks to the Author):

SUMMARY

This manuscript uses tridimensional imaging techniques of cleared tissues to study implantation in the mouse. The results support the idea that the implantation chamber for crypt is formed by epithelial invaginations that arise from the uterine lumen which has preexisting glands. Further, epithelial-specific deletion of Vangl2, a core component of the planar cell polarity pathway, found that pathway to be critical for implantation crypt formation as well as embryo spacing and spatial relationships with endometrial glands. The results provide evidence that crypt architecture and direct communication with glands is a previously unrecognized fundamental step requisite for implantation and pregnancy success.

OVERALL AND MAJOR COMMENTS

This manuscript contains some spectacular histomorphological data on how implantation proceeds in a normal and disrupted pregnancy using mice as a model system. The studies are novel and very well conducted, and the data fully support the major claims that glandular architecture and communication is essential for pregnancy establishment.

Major Comments:

- (1) Gene nomenclature needs to be consistent, such as PGR for progesterone receptor and ESR1 for estrogen receptor alpha, CTNNB1 for beta-catenin, CDH1 for E-cadherin and so forth. Adherence to MGI nomenclature is suggested.
- (2) Figure 4h: Provide more specific details on the co-immunoprecipitation of FOXA2 and ESR1. What day of pregnancy was the uteri used for the experiment? This experiment seems to be an add-on that could be removed from the manuscript as it is a bit tangential and does not appreciably add much.
- (3) This manuscript provides technical and conceptual advances in pregnancy biology and will significantly impact the field. Given the technical nature of the paper, it would be good to have the following details for reproducibility: (a) Table for the antibodies including their source and conditions of the primary and secondary antibodies; (b) a very detailed protocol for the tissue clearing, immunostaining, and light sheet and two photon microscopy as well as downstream software analysis. These items could be provided as supplementary information.

SPECIFIC COMMENTS

Line Comment

23 The uterine glands likely secrete factors in an exocrine manner (toward lumen) and paracrine manner (toward stroma)

54 Define Vangl2 briefly

60 PR should be PGR

142 Replace drain with direct

172 Consider replacing noodles with pasta

193 Please provide the "data not shown"

213 The use of display is awkward
455 delipidation should be dilipidated
260 Typically the Cre is placed before the floxed gene
Fig. 4 Analysis is not spelled correctly in legend

Reviewer #2 (Remarks to the Author):

The manuscript by Yuan and coworker describes the use of novel 3D imaging to analyze the uterine gland topography and crosstalk during implantation. This is conducted on wild type mice and mice in which the *Vangl2* gene was ablated in the uterine epithelium with the *Ltfc* model. The ablation of *Vangl2* in the uterus has been conducted using the *PRcre* and the phenotype is a sever pregnancy defect. The use of the *Ltf cre* results in a hypomorphic phenotype with which imaging can be conducted. Using a hypomorphic mouse model is an excellent means of demonstrating how this approached allows visualization of the interactions of the glandular epithelium using this technology. This manuscript then does an excellent job showing how this approach adds new incite to the biology of the uterine epithelium during embryo. However, there are several concerns.

The Manuscript states that the difference could be due to the *PRCre* model which ablates genes in the neonatal uterus and deletes one copy of the *PR* allele. The hypomorphic phenotype is attributed to either loss of one allele of the *PR* gene or developmental reprogramming. Although the latter may be correct, loss of one all of *PR* does not affect pregnancy. The investigators should also address the issue that *Vangl2* may be acting in compartments other than the uterine epithelium or that ablation using the *Ltf cre* may be incomplete or delayed. Their supplemental data analyzing the ablation indicates that this is most likely the case (Extended data figure 1). This should be addressed

The approach in this manuscript should be compared to the previous report by Arora et al. Development 2016 that used the similar technology to describe implantation in the mouse uterus. This would highlight the novelty of this manuscript.

The manuscript clearly needs to highlight the uniqueness of this hypomorphic phenotype or how this approach adds new information to the literature since both the phenotype and approach have been previously published.

Reviewer #3 (Remarks to the Author):

In this manuscript, Yuan et al., investigates embryo-uterine interactions to study pregnancy success using tissue clearing technology. They check crypt-gland topography in epithelial-specific deletion of *Vangl2* at different times (mice perfused at different time points for tissue clearing study) to show that it plays a critical in the correct formation of crypt-gland structures. Overall, the integration of clearing technology seems very nice and informative, but I have concerns related to novelty of the study: what we learn more from 3D images (compared to a traditional though histology) and from the new *Vangl2* conditional KO (deleting it even in a smaller subset) compared to previously generated *Vangl2* KOs.

Overall, I believe the authors should dive more into generating new biological information using the modern technology with the quantification going beyond the state-of-the-art. My detailed comments in this line are as below:

1) The authors previously published a paper using a conditional KO mouse in which Vangl2 was deleted in progesterone receptor cells. They obtained convincing data showing that Vangl2 was critical to pregnancy success. The uterine epithelium expresses progesterone receptor, so basically it seems to that here they also remove Vangl2 in the same compartment as last time, and with the same defects in pregnancy rate success. Therefore, it is not clear to me how much more we learn with this new transgenic line, in which they removed Vangl2 in the uterine epithelium.

2) I still wonder where is the direct confirmation that alterations in the glandular 3D organization could be responsible for the miscarriage? Normally, during a miscarriage, there's an increase in the inflammation in the uterus which starts secreting toxic molecules for the embryos. The authors seem even not to look at markers of inflammation (ideally in 3D).

3) Could the authors measure the glandular secretion and compare wildtype and KO mice? This could be another evidence to justify that the embryos die in utero.

4) There seems to be no great difference between the light-sheet and two-photon images nor a clear justification for performing two kinds of acquisitions. The rationale beyond these technical efforts remains very vague.

5) I am also not very convinced what is the advantage of 3D imaging in the study- for example in Figure 2, could similar images/results be obtained with standard serial histology? It would be important to see comparative traditional histology images and judge how much more we learn from clearing-based 3D images.

6) I miss quantifications, statistics and number of mice in many of the figures related to tissue clearing, where the major biological findings were intended to be presented e.g., Figures 2-4. Observed defects on a single mouse could be very well due to a technical problem such as in faulty staining, clearing or imaging.

7) The tdTomato signal is not well preserved by DISCO clearing. Therefore, the authors seem to only PFA fix Rosa26tdTomatoLtfCre/+ reporter and do a brief clearing in RIMS solution. These brief clearing results seem not very different than DISCO cleared samples. Therefore, I again question if similar results could not be obtained with traditional histology on such relatively small biological samples. In addition, please express very clearly which tissue is cleared how (if different protocols are used) and also comment on signal preservation of tdTomato in your specific organic-solvent based clearing.

8) It is not clear which imaging parameters are used for each figure/video: please indicate microscope, objective, image acquisition parameters, and ~time. Please indicate what the

pseudo colors in the videos are. Please mark the point of interest/s in the videos with an arrow etc., that reader can focus and see what biological changes are meant to be seen.

9) It is not clear how the authors performed image segmentation. In general, commercial softwares are poor in recognizing specific shapes. They mainly rely on the signal intensity. For example, how the decision is made in Figure 2h, what is lumen and what is E-cad signal from the original image in Figure 2f if there is no double labeling?

10) Organic solvent-based DISCO clearing methods shrink the tissue. Could authors comment on what is the shrinkage rate in uteri? Related to that, did the authors re-adjusted their scale bars throughout the manuscript by considering the shrinkage rate?

Reviewers' Comments

Reviewer 1 (Remarks to the Author):

We are very pleased with this reviewer's positive comments. We have addressed all of the minor suggestions offered by this reviewer.

SUMMARY

This manuscript uses tridimensional imaging techniques of cleared tissues to study implantation in the mouse. The results support the idea that the implantation chamber for crypt is formed by epithelial invaginations that arise from the uterine lumen which has preexisting glands. Further, epithelial-specific deletion of *Vangl2*, a core component of the planar cell polarity pathway, found that pathway to be critical for implantation crypt formation as well as embryo spacing and spatial relationships with endometrial glands. The results provide evidence that crypt architecture and direct communication with glands is a previously unrecognized fundamental step requisite for implantation and pregnancy success.

OVERALL AND MAJOR COMMENTS

This manuscript contains some spectacular histomorphological data on how implantation proceeds in a normal and disrupted pregnancy using mice as a model system. The studies are novel and very well conducted, and the data fully support the major claims that glandular architecture and communication is essential for pregnancy establishment.

Major Comments:

(1) Gene nomenclature needs to be consistent, such as PGR for progesterone receptor and ESR1 for estrogen receptor alpha, CTNNB1 for beta-catenin, CDH1 for E-cadherin and so forth. Adherence to MGI nomenclature is suggested.

All gene nomenclature is now consistent throughout the manuscript.

(2) Figure 4h: Provide more specific details on the co-immunoprecipitation of FOXA2 and ESR1. What day of pregnancy was the uteri used for the experiment? This experiment seems to be an add-on that could be removed from the manuscript as it is a bit tangential and does not appreciably add much.

ESR1 and FOXA2 are primarily present in glands and have been shown to cooperate in gland function. We did this experiment in day 4 uterus to show their physical interaction which has been detected previously in the liver (Cell. 2012; 148(1-2): 72–83). These results are now presented in a supplemental figure (Extended Figure. 7).

(3) This manuscript provides technical and conceptual advances in pregnancy biology and will significantly impact the field. Given the technical nature of the paper, it would be good to have the following details for reproducibility: (a) Table for the antibodies including their source and conditions of the primary and secondary antibodies; (b) a very detailed protocol for the tissue clearing, immunostaining, and light sheet and two photon microscopy as well as downstream software analysis. These items could be provided as supplementary information.

These are very helpful suggestions and we have included all of the requested information in the revised manuscript with a web link of the downstream software analysis.

SPECIFIC COMMENTS

Line Comment

23 The uterine glands likely secrete factors in an exocrine manner (toward lumen) and paracrine manner (toward stroma)

This is an interesting point, but needs confirmation by experimental evidence.

54 Define Vangl2 briefly

Vangl2 is an abbreviation of Van Gogh-Like Protein 2, a core component of planar cell polarity (PCP) signaling. It is now stated in the revised version.

60 PR should be PGR

PR is replaced by PGR

142 Replace drain with direct

This is now corrected.

172 Consider replacing noodles with pasta

This change is now made in the revised draft.

193 Please provide the “data not shown”

The data is now shown in a supplemental figure (Extended Figure. 7).

213 The use of display is awkward

We have replaced ‘display’ with ‘landscape’.

455 delipidation should be dilipidated

This is now corrected as suggested.

260 Typically the Cre is placed before the floxed gene

We feel that either way should be acceptable nomenclature if appropriately defined.

Fig. 4 Analysis is not spelled correctly in legend

We have corrected this typographical error.

Reviewer 2 (Remarks to the Author)

We are also pleased with the comments of this reviewer and have addressed the concerns raised by this reviewer.

The manuscript by Yuan and coworker describes the use of novel 3D imaging to analyze the uterine gland topography and crosstalk during implantation. This is conducted on wild type mice and mice in which the Vangl2 gene was ablated in the uterine epithelium with the Ltfcre model. The ablation of Vangl2 in the uterus has been conducted using the PRcre and the phenotype is a sever pregnancy defect. The use of the Ltf cre results in a hypomorphic phenotype with which imaging can be conducted. Using a hypomorphic mouse model is an excellent means of demonstrating how this approached allows visualization of the interactions of the glandular epithelium using this technology.

This manuscript then does an excellent job showing how this approach adds new incite to the biology of the uterine epithelium during embryo. However, there are several concerns.

The Manuscript states that the difference could be due to the PRCre model which ablates genes in the neonatal uterus and deletes one copy of the PR allele. The hypomorphic phenotype is attributed to either loss of one allele of the PR gene or developmental reprogramming. Although the latter may be correct, loss of one all of PR does not affect pregnancy.

Yes, it has been shown that loss of one PR allele does not appear to affect pregnancy outcomes. However, it is still debated that one cannot unequivocally exclude subtle changes that occur in various tissues that express PGR. Nonetheless, our primary objective was to see which uterine cell types (myometrium, stroma or epithelium) contribute to Vangl2 deficiency and pregnancy success, since Vangl2 is expressed in both epithelial and stromal cells, albeit at low levels in the stroma. That is why we utilized *Ltf-Cre* line to specifically delete epithelial Vangl2 and assess the process of implantation and pregnancy outcomes.

The investigators should also address the issue that Vangl2 may be acting in compartments other than the uterine epithelium or that ablation using the *Ltf-Cre* may be incomplete or delayed. Their supplemental data analyzing the ablation indicates that this is most likely the case (Extended data figure 1). This should be addressed.

It is a legitimate comment. We performed experiments in whole tissue extracts, although immunofluorescence (IF) showed near complete deletion of Vangl2 in the epithelium with intact expression in stromal cells. The logic behind using *Ltf-Cre* mice is to dissect the function of Vangl2 in the epithelial cells versus stromal cells. To confirm the deletion of Vangl2 in epithelial cells, we now separated epithelial cells from stromal cells from both *Vangl2^{ff}* and *Vangl2^{ff}Ltf^{Cre/+}* mice, and qRT-PCR was performed in these separated cells. The results show clearly that Vangl2 is dramatically deleted (>99% deletion) in *Vangl2^{ff}Ltf^{Cre/+}* epithelial cells with retention of Vangl2 in floxed and deleted stromal cells. We strongly feel that it is imperative to investigate the function of epithelial Vangl2 to explore its function in epithelial evagination and crypt formation in the presence of an active blastocyst, since epithelial cells expressing Vangl2 physically are in close apposition with the blastocyst, although the role of stromal Vangl2 cannot be completely ruled out.

The approach in this manuscript should be compared to the previous report by Arora et al. Development 2016 that used the similar technology to describe implantation in the mouse uterus. This would highlight the novelty of this manuscript. The manuscript clearly needs to highlight the uniqueness of this hypomorphic phenotype or how this approach adds new information to the literature since both the phenotype and approach have been previously published.

This is a legitimate question. The work by Arora et al. used mouse uterus to develop the technique of 3D imaging after antibody staining. In fact, we provided a substantial number of mouse uteri for their studies, as evident in their Acknowledgements section of the Development paper. Their work primarily shows gland orientation with respect to the location of the embryo only on day 5 morning. Our study illuminates the role of *Vangl2* in various steps in the complex and dynamic process of implantation in time.

We have included substantial additional new data in the revised manuscript to illustrate the expanded scope our work. We show here a novel physiologic finding that crypts (implantation chambers) are not only form by directed, evenly spaced projections, but also that they form in conjunction with preexisting glands to confer direct access of the embryo to gland secretion within the crypt. Furthermore, using diapausing mice conferring embryonic dormancy in floxed mice and mice with *Vangl2* deletion exclusively in uterine epithelial cells, we show that dynamic changes in gland topography depend on implantation-competent (activated) blastocysts and *Vangl2*/PCP (planar cell polarity). By transferring blastocyst-size beads preloaded with HB-EGF in pseudopregnant mice, we found that HB-EGF signaling is a molecular interface between the blastocyst and crypt epithelium downstream of *Vangl2*/PCP signaling to trigger crypt formation and facilitate communication between embryos and glands. Glands that directly connect the crypt encasing the embryo during implantation were not previously recognized by two dimensional studies. Our study revealed the crypt-gland topography and the fundamental role this relationship plays in pregnancy success.

Reviewer 3 (Remarks to the Author)

In this manuscript, Yuan et al., investigates embryo-uterine interactions to study pregnancy success using tissue clearing technology. They check crypt-gland topography in epithelial-specific deletion of *Vangl2* at different times (mice perfused at different time points for tissue clearing study) to show that it plays a critical in the correct formation of crypt-gland structures.

Overall, the integration of clearing technology seems very nice and informative, but I have concerns related to novelty of the study: what we learn more from 3D images (compared to a traditional though histology) and from the new Vangl2 conditional KO (deleting it even in a smaller subset) compared to previously generated Vangl2 KOs. Overall, I believe the authors should dive more into generating new biological information using the modern technology with the quantification going beyond the state-of-the-art. My detailed comments in this line are as below:

We appreciate this reviewer's comments and address them below.

Traditional serial sectioning to obtain similar uterine topography showing crypt formation and direct gland-embryo communication within the crypt in time would be a Herculean method of obtaining meaningful information considering the number of implantation sites, number of animals and different treatment groups to be assessed. In contrast, 3D visualization of the whole uterus with the initiation and progression of the implantation process in time by deep tissue clearing and imaging generates a global view of the process more accurately and promptly, with intact anatomic positions of related structures. The unique feature of the process of implantation and new biological insights with respect to implantation are already stated in response to the Reviewer 2.

1) The authors previously published a paper using a conditional KO mouse in which Vangl2 was deleted in progesterone receptor cells. They obtained convincing data showing that Vangl2 was critical to pregnancy success. The uterine epithelium expresses progesterone receptor, so basically it seems to that here they also remove Vangl2 in the same compartment as last time, and with the same defects in pregnancy rate success. Therefore, it is not clear to me how much more we learn with this new transgenic line, in which they removed Vangl2 in the uterine epithelium.

Our responses to this comments are already elaborated in Reviewer 2's comments.

2) I still wonder where is the direct confirmation that alterations in the glandular 3D organization could be responsible for the miscarriage? Normally, during a miscarriage, there's an increase in the inflammation in the uterus which starts secreting toxic molecules for the embryos. The authors seem even not to look at markers of inflammation (ideally in 3D).

In our opinion, we did not observe expulsion of fetuses from the uterus, normally termed miscarriages. Rather we found resorptions, which are often seen in mice with defective implantation. It is still an enigma how a few fetuses can reach term pregnancy with healthy delivery while neighboring fetuses undergo resorptions in the same uterine horn. Of course, there could be some inflammation in the uterus with resorbing fetuses at later stages of pregnancy. However, our main objective is to define the events surrounding early stages of pregnancy, namely implantation and decidualization. Exploration of the markers of inflammation is not within the scope of the present study.

3) Could the authors measure the glandular secretion and compare wildtype and KO mice? This could be another evidence to justify that the embryos die in utero.

This is an interesting point which was pursued by many investigators for decades. While there are some attempts to measure the component of uterine fluids by Mass Spec, the reliability of the findings are questionable. The luminal volume of each uterine horn is only between 150-200 nL in mice. Therefore, flushing uterine horns with 300-500 μ L will result in damage of the luminal lining and the flushing will be contaminated with epithelial tissue components, limiting comparison of glandular secretion between WT and KO mice (J Reprod Fertil. 1981, 62(1):105-109; J Reprod Fertil. 1984, 71(1):73-80). Thus, this procedure is rarely practiced in mouse implantation studies.

4) There seems to be no great difference between the light-sheet and two-photon images nor a clear justification for performing two kinds of acquisitions. The rationale beyond these technical efforts remains very vague.

This is a reasonable comment. To clarify, the main purpose of our use of light-sheet microscopy was to confirm our two-photon results. The light-sheet platform can image larger tissue rapidly but generates a huge amount of data, which is a challenge for data analysis. The multi-photon microscope can image different stages of the whole uterus and is compatible with different clearing reagents.

5) I am also not very convinced what is the advantage of 3D imaging in the study- for example in Figure 2, could similar images/results be obtained with standard serial histology? It would be

important to see comparative traditional histology images and judge how much more we learn from clearing-based 3D images.

Both 2D and 3D imaging platforms have their own advantages and shortcomings. Traditional sectioning of the uterus is normally performed for molecular and cell biological information, because immunolocalization and in situ hybridization are easily applicable. 3D visualization is sometimes limited by poor antibody penetration and staining; this requires the use of different fixatives and in situ hybridization is not yet possible for the 3D platform. On the other hand, 3D imaging provides a global view of the uterine topography during implantation in time to capture the dynamic nature and progression of this process in time. That is why, we used both traditional sectioning and 3D visualization to generate a comprehensive understanding of the implantation process.

6) I miss quantifications, statistics and number of mice in many of the figures related to tissue clearing, where the major biological findings were intended to be presented e.g., Figures 2-4. Observed defects on a single mouse could be very well due to a technical problem such as in faulty staining, clearing or imaging.

The number of animals are included in Experimental Procedure Section. Each experiment was performed in at least three mice for each experimental group.

7) The tdTomato signal is not well preserved by DISCO clearing. Therefore, the authors seem to only PFA fix Rosa26tdTomatoLtfCre/+ reporter and do a brief clearing in RIMS solution. These brief clearing results seem not very different than DISCO cleared samples. Therefore, I again question if similar results could not be obtained with traditional histology on such relatively small biological samples. In addition, please express very clearly which tissue is cleared how (if different protocols are used) and also comment on signal preservation of tdTomato in your specific organic-solvent based clearing.

Currently, both the DISCO staining protocol and RIMS clearing are necessary to exhibit the full picture of whole uterus. The major difference is that PFA fix will cause poor antibody penetration and DISCO staining will quench tdTomato fluorescence since the DISCO protocol requires treatment with methanol. This is the reason why we used two different clearing methods in our studies. All DISCO-cleared tissues are indicated in the figures for each antibody and RIMS-cleared tissues are labeled as tdTomato in the figures. We checked PFA fixed tdTomato tissue

before and after RIMS clearing under stereomicroscopy and found that the tdTomato signal become much sharper and brighter after clearing in RIMS for 3 days and the signal is retained even 6 months later in 4°C. Notably, the signal is totally bleached after fixing in Methanol:DMSO mixture for 10 min. Other than RIMS, the use of THF (tetrahydrofuran) and DCM/DBE (dichloromethane/ dibenzyl ether) will also cause loss of tdTomato signal to some extent even fixed in PFA. Therefore, we only use RIMS for clearing of tdTomato uteri.

8) It is not clear which imaging parameters are used for each figure/video: please indicate microscope, objective, image acquisition parameters, and ~time. Please indicate what the pseudo colors in the videos are. Please mark the point of interest/s in the videos with an arrow etc., that reader can focus and see what biological changes are meant to be seen.

Microscope, objective and image acquisition are included in the revised version. The pseudocolours are also annotated in the videos with appropriate labeling to make the video more meaningful and readable.

9) It is not clear how the authors performed image segmentation. In general, commercial softwares are poor in recognizing specific shapes. They mainly rely on the signal intensity. For example, how the decision is made in Figure 2h, what is lumen and what is E-cad signal from the original image in Figure 2f if there is no double labeling?

Imaris (Version 8.3) was employed for image segmentation under the instruction of Alain Chedotal (Cell 2017 Mar 23;169(1):161-173.) using single antibody staining. We also performed double antibody staining of CK8 and FOXA2 to confirm this segmentation. The detailed protocol for segmentation is provided with a web link. There is a tutorial in the official website of bitplane (Imaris) on how to do this segmentation: <http://www.bitplane.com/learning/3d-reconstruction-and-segmentation-of-whole-slide-images-an-expert-imagis-tip>

1. Select the "edit"-Tab of a surface object
2. In order to execute a vertical cut, rotate the camera and tilt the connected objects into a horizontal orientation.
3. "shift-click" with the 3D-Cursor at the position of the cut (a yellow line appears), Rotate the camera in order to see if the cut fits.

4. Press the "Cut" Button from the edit tab (the object splits into two individual surface objects).

10) Organic solvent-based DISCO clearing methods shrink the tissue. Could authors comment on what is the shrinkage rate in uteri? Related to that, did the authors re-adjusted their scale bars throughout the manuscript by considering the shrinkage rate?

The shrinkage rate of different clearing methods is addressed in this Cell snapshot (Cell. 2017;171(2):496-496.) which clearly states shrinkage rate is minimal. We did not adjust the scale bar according to the shrinkage rate, but both controls and experiments were similarly treated. Notably, shrinkage also occur in varying degrees in traditional histological sectioning depending on the fixative used. However, traditionally the shrinkage correction is not used for scale bars in 2D imaging.

REVIEWERS' COMMENTS:

Reviewer #2 (Remarks to the Author):

The authors have adequately addressed the issues in this manuscript and it is acceptable for publication.

Reviewer #3 (Remarks to the Author):

My comments to Revision

My remaining comments are as below- if the authors could address, they could improve the paper some more. I would not need to see manuscript again.

-To simplify the reading, please carry the technical info on microscope setup and clearing as a table to supplemental info.

-The authors response 9- I take this response as we do not know. It is well accepted by the field that there is no software to "reliably" analyze the large 3D data. In particular, commercial software generating data based on thresholding such as Imaris cannot be alone reliable, unless it is confirmed by independent method on the same dataset. I found a bit enigmatic that the authors refer to user manual of a commercial software to justify their results instead of addressing what have been ask (For example, how the decision is made in Figure 2h, what is lumen and what is E-cad signal from the original image in Figure 2f if there is no double labeling?). Needless to say, learning it from an expert from the field does not make the conducted results accurate or confirmed. Therefore, the authors should clearly state this weakness in the main text such as "here we used a commercial software to segment the data and in the future development of more reliable software e.g., based on machine-learning will yield better ways to view and interpret the 3D data..."

-The authors response 10 - Belle et al., 2017 Cell paper, which is about human embryos (a different tissue) refers to another paper to give shrinkage rate of mouse tissue in DISCO clearing. Here, the authors address my query by referring to a paper, which refers to a paper instead of performing some simple before and after clearing measurements for mouse uteri. Please check the literature more carefully and refer to correct/original papers when information is given from literature only.

REVIEWERS' COMMENTS:

Reviewer #2 (Remarks to the Author):

The authors have adequately addressed the issues in this manuscript and it is acceptable for publication.

Reviewer #3 (Remarks to the Author):

My comments to Revision

My remaining comments are as below- if the authors could address, they could improve the paper some more. I would not need to see manuscript again.

-To simplify the reading, please carry the technical info on microscope setup and clearing as a table to supplemental info.

This information is included in both the figure legends and Method.

-The authors response 9- I take this response as we do not know. It is well accepted by the field that there is no software to "reliably" analyze the large 3D data. In particular, commercial software generating data based on thresholding such as Imaris cannot be alone reliable, unless it is confirmed by independent method on the same dataset. I found a bit enigmatic that the authors refer to user manual of a commercial software to justify their results instead if

addressing what have been ask (For example, how the decision is made in Figure 2h, what is lumen and what is E-cad signal from the original image in Figure 2f if there is no double labeling?). Needless to say, learning it from an expert from the field does not make the conducted results accurate or confirmed. Therefore, the authors should clearly state this weakness in the main text such as "here we used a commercial software to segment the data and in the future development of more reliable software e.g., based on machine-learning will yield better ways to view and interpret the 3D data..."

We understand that a commercial software may not be totally compatible for 3D image analysis for segmentation. Therefore, to confirm our segmentation data using Imaris, we performed single and double immunostaining of tissues with CK8, E-cadherin (E-cad) and FOXA2 to separate glands and lumens as applicable. Double immunostaining with CK8 and FOXA2 (gland specific) was included in the revised manuscript (Extended Fig. 4, (New Figure 4)). Pseudocoloring was applied to separate glands and lumen primarily for the ease of visualization.

-The authors response 10 - Belle et al., 2017 Cell paper, which is about human embryos (a different tissue) refers to another paper to give shrinkage rate of mouse tissue in DISCO clearing. Here, the authors address my query by referring to a paper, which refers to a paper instead of performing some simple before and after clearing measurements for mouse uteri. Please check the literature more carefully and refer to correct/original papers when information is given from literature only.

We addressed this issue during the last revision. As we stated that normally tissue shrinkage occurs in both 2D and 3D tissue analysis. Notably, we performed 3D analysis of control and mutant tissue under similar conditions.